# Personality, Preferences, Satisfaction, and Achievement in a Biostatistics Course: Traditional versus Flipped Classrooms in Nursing Education

**Aina M. Yañez** [1,2] , **Daniel Adrover-Roig** [3,4,*] **and Miquel Bennasar-Veny** [1,2,5]

1  Department of Nursing and Physiotherapy, University of Balearic Islands, 07122 Palma, Spain
2  Health Research Institute of the Balearic Islands (IdISBa), 07120 Palma, Spain
3  Department of Applied Pedagogy and Educational Psychology, University of Balearic Islands, 07122 Palma, Spain
4  Institute of Research and Innovation in Education (IRIE), University of Balearic Islands, 07122 Palma, Spain
5  CIBER de Epidemiología y Salud Pública (CIBERESP), 28029 Madrid, Spain
*  Correspondence: daniel.adrover@uib.es; Tel.: +34-971172426

**Abstract:** Background: In higher education, there is a lack of knowledge about how individual factors influence personal preferences for teaching methodology and academic outcomes in biostatistics among nursing students. This study sought to evaluate the associations between personality traits and preferences for the flipped classroom or traditional lessons, satisfaction, and achievement in a biostatistics course in nursing education. Methods: The cross-sectional study included data collection at two time points. During the 2018–2019 academic course, the flipped classroom approach was implemented into a biostatistics course of the nursing degree at the University of the Balearic Islands (Spain). Students responded to an online questionnaire including sociodemographic data, personality traits, locus of control, satisfaction, teaching quality, student involvement, and effort. After the final examination, students' achievements were collected. Results: A total of 137/161 (85.1%) students were included in the study. Most students preferred a flipped classroom to traditional lectures. Students who scored high in neuroticism preferred the traditional class ($p < 0.001$). Furthermore, personality was associated with satisfaction: high levels of responsibility and internal locus of control were significantly associated with higher satisfaction ($p < 0.05$). The percentage of students who did not pass the biostatistics exam was reduced by 50% with the flipped classroom methodology (33% vs. 15%; $p < 0.001$). Conclusions: A flipped classroom is adequate for the biostatistics course. Teachers should reconsider possible reservations about new teaching methods, which can be related to student personality.

**Keywords:** flipped classroom; traditional lecture; personality traits; nursing; education research

## 1. Introduction

The flipped classroom (FC) approach is a type of educational method in which the standard lecture and homework elements are "flipped" [1]. The FC changes the traditional model (lecture-based classroom) by encouraging students to actively engage in learning [2]. Specifically, students view the lecture content at home using tutorials, videos, and documents prior to the class, including pre-recorded lectures. Once in class, their time is dedicated to engaging in student-centered learning activities, such as workgroups, problem solving, or interactive activities. Furthermore, an FC allows students to review course-related materials after the class [3,4].

FCs have gained increasing interest across health sciences in higher education, and its use is continuously growing [5–8]. An FC could be a suitable teaching strategy to prepare the healthcare workforce because it improves learning, encourages participation, and improves clinical reasoning skills [2,9]. This learning method might be of particular use

in assisting nursing students with subjects considered difficult, such as biostatistics, thus possibly enhancing the understanding of its applicability. For example, the FC approach could be beneficial in a biostatistics course as it offers a high degree of flexibility and promotes self-regulated learning through practical exercises and problem solving using real data [10,11]. However, there is currently a lack of studies evaluating the effectiveness of an FC in the subject of biostatistics.

Although the literature supports the effectiveness of an FC in higher education nursing programs and an improvement in student satisfaction with this methodology, not all students benefit equally [1,12]. Some students were satisfied with the FC approach, but their test scores did not improve, while others revealed that students were less enthusiastic but achieved higher scores [13].

Therefore, individual factors may influence the preference for a given educational methodology and academic performance; however, these are still not well understood. Several studies have found that individual factors such as the locus of control, academic resilience, motivation, and personality traits, could be associated with academic performance among university students, especially in health sciences [14–16]. In this vein, previous studies have highlighted that class attendance is associated with variables that reflect high levels of self-discipline (e.g., high conscientiousness) and a sense of internal control (internal locus of control) over academic achievement. Many of these variables also show significant relationships with academic outcomes [17,18]. More specifically, locus of control refers to the explanations that people provide for the causes of their behaviors, and the psychological terms aiming to explain these processes are referred to as attributions. In psychology, these attributions help to better understand human motivation and sense of competence.

For instance, participants who score high on internal locus of control tend to attribute responsibility for their actions to themselves, whereas those scoring high on external locus of control explains their successes and failures to external factors, such as luck or coincidence [17]. Additionally, an internal locus of control tends to be directly associated with some dimensions of personality, such as conscientiousness [19], which is one of the most powerful non-cognitive predictors of academic achievement [20]. In this regard, we departed from the Big Five model of personality [21], which includes the dimensions neuroticism (emotional instability), extraversion, conscientiousness, agreeableness, and openness to experience. As stated, we were particularly interested (but not exclusively) in exploring the role of conscientiousness, which is the tendency to carry out a task efficiently and carefully and to take obligations seriously. People scoring high in conscientiousness tend be organized and tend to show behaviors of self-discipline and to aim for achievement [22].

A recent systematic review reported inconsistent associations between individual (non-cognitive) factors and academic outcomes among students of health professions; however, higher conscientiousness, academic resilience, and grit were generally associated with better outcomes [23]. Nevertheless, there is a knowledge gap about how individual factors, such as locus of control or personality dimensions, are associated with personal preferences for a given teaching methodology and academic outcomes among nursing students. Thus, we aimed to consider both potentially relevant variables regarding the FC model since both seem to play roles in discipline and commitment.

The main aim of this study was to evaluate the association between personality traits, teaching method preference, satisfaction, and academic achievement in a biostatistics course in nursing education. We also compared the academic performance with that of the previous year, in which a more conventional teaching methodology was used.

## 2. Materials and Methods

We addressed the following research questions: (1) How are the Big Five personality traits associated with satisfaction, academic achievement, and preference for a given teaching methodology? (2) How is the locus of control associated with satisfaction, academic

achievement, and preference for a given teaching methodology? (3) What is the impact of a flipped class on the academic performance of students? We performed a cross-sectional study which included two data-collection time points. We collected data on sociodemographic, personality dimensions, and locus of control during an initial assessment. In addition, data on preferences, satisfaction, and achievement were collected during the final assessment.

This study was conducted in the Nursing and Physiotherapy Faculty of the University of the Balearic Islands in Spain. A total of 161 nursing degree students enrolled in the Statistics for Health Sciences course during the 2018–2019 academic year. Of those, 137 (85%) agreed to participate and signed informed consent. The mean age was 22.4 ± 7 years, 93.4% of participants were Caucasian, and 79.6% were women. Most students were upper secondary education (49%) and advanced vocational training (34%) graduates who passed the university admission exam; the remaining students gained accessed through special access for people over 25 years (9%), or as second-degree students or through a university degree change (8%).

Students who agreed to participate signed informed written consent. The study was approved by the ethics research committee (CER) of the University of the Balearic Islands (CER76/18).

## 2.1. Implementation of the Flipped Classroom

Before integrating the FC into the course, the teacher, mentored by experts, reviewed all resources and tools to be implemented in the educational intervention in biostatistics. Students were required to read the material provided and to watch each session's tutorials and videos before attending the class. Class time included group work, problem-solving, quizzes, etc. Each class started with an interactive online quiz designed to assess student learning and to address possible difficulties with the topic. Students were then divided into small groups (5–6 students) and asked to solve a series of problems while the teacher provided feedback to help building meaningful learning (Table 1).

**Table 1.** Comparative analysis of flipped and traditional classroom for a biostatistics course.

|  | Flipped | Traditional |
|---|---|---|
| Pre-classroom | Students are introduced to the topic and given study material as videos, material to read and understand. | Learning program schedules are provided |
| Attendance in classroom | Online quiz (Kahoot platform) Brief review of study material Additional explanations Presentation of questions and problems to be discussed in group | Lectured lessons Students watch and listen and take notes |
| Post-classroom | Individual discussion activities | Individual problem solving (homework) Study materials are given |

## 2.2. Data Collection

Data were collected at two different time points. At baseline (on the first day of class), students were asked to respond to an online questionnaire that included sociodemographic data, the NEO-Five Factor Inventory (NEO-FFI) [21], and the bi-dimensional locus of control scale [24]. The NEO-FFI evaluates, with 60 Likert-type items (from 'completely disagree' to 'completely agree'), the main personality domains of the five-factor model: neuroticism (tendency to experience negative affect, such as sadness, anxiety, guilt, or impulsivity), extraversion (tendency toward assertiveness, a high level of energy, sociability, and optimism), openness (tendency for new experiences, intellectually curious, and aesthetically sensitive), agreeableness (tendency toward trust, cooperation, and altruism), and conscientiousness (tendency toward self-control, organization, motivation, and reliability).

For the present study, direct scores for each domain were calculated. The NEO-FFI has been widely used in clinical and basic research studies. Its internal consistency ranges from 0.86 to 0.95, and the retest reliabilities range from 0.66 to 0.92 across various samples. The bi-dimensional locus of control (LOC) scale [24] measures attribution styles and includes 30 items, of which 13 correspond to an external locus of control (scores range from 0 to 52 points) and 10 correspond to an internal locus of control (scores range from 0 to 40 points). Internal consistency reliabilities are high for both the external LOC subscale, 0.85, and for the internal LOC subscale, 0.87.

On the last day of the course, students were asked to respond to a second online survey using five-answer options Likert scale (from 1 = unsatisfied to 5 = very satisfied) to evaluate course satisfaction, including lecture structure, organization workload, promotion of student's engagement (enthusiasm and participation), and the utility of video lectures. Students were also asked to score their effort levels (low, medium, high, and very high), to score their learning level (low, medium, high, and very high), to express their preferred teaching method (traditional, FC, or indifferent), and to note their adherence to scheduled homework tasks (tutorials, videos, and documents including pre-recorded lectures) to be completed before the class session (Yes, at least 80% of the content; No, I do not have enough time; and No, I forgot).

Finally, after the final examination, students' marks for the 2018–2019 current academic course were compared with those of the previous course (2017–2018), retrieved using computerized registers. Group characteristics, learning objectives, and evaluation were similar for both cohorts although the learning tasks were different (Table 1).

### 2.3. Data Analysis

Quantitative descriptive variables are expressed as means ($\pm SD$s), and differences according to group preference were analyzed with Student's *t*-test when normal distribution was assumed. The normal distribution was assessed for the personality dimensions by the Kolmogorov–Smirnov test. The categorical variables are expressed as $n$ (%) and were compared using the chi-square ($\chi^2$) statistic. Spearman correlations (*Rho*) were calculated to evaluate the association between personality traits, locus of control, and the level of satisfaction with the FC method, given that satisfaction variables were ordinal. Cohen's *d* was used to interpret effect sizes and correlation coefficients [25].

The analyses were performed using the Statistical Package for the Social Sciences (SPSS) software version 25.0 (IBM Company, New York, NY, USA) for Windows. The statistical tests were two-sided, and *p* values < 0.05 were considered statistically significant.

### 3. Results

Students positively valued the FC method, with 41% preferring FC over other methods, 23% preferring traditional class, and the remaining 36% not showing any preference (Figure 1). Specifically, students' satisfaction with FC was high in all the measured aspects: in terms of enthusiasm (4.6/5 $\pm$ 0.7), the utility of the video lectures (4.6 $\pm$ 0.6), students' participation (4.4/5 $\pm$ 0.7), course organization (4.2/5 $\pm$ 0.6), and workload (4.1 $\pm$ 0.6).

Most of the students (84%) declared a high adherence to the scheduled homework tasks to be completed before the class session (at least 80% of the content) as well as a high satisfaction with the workload.

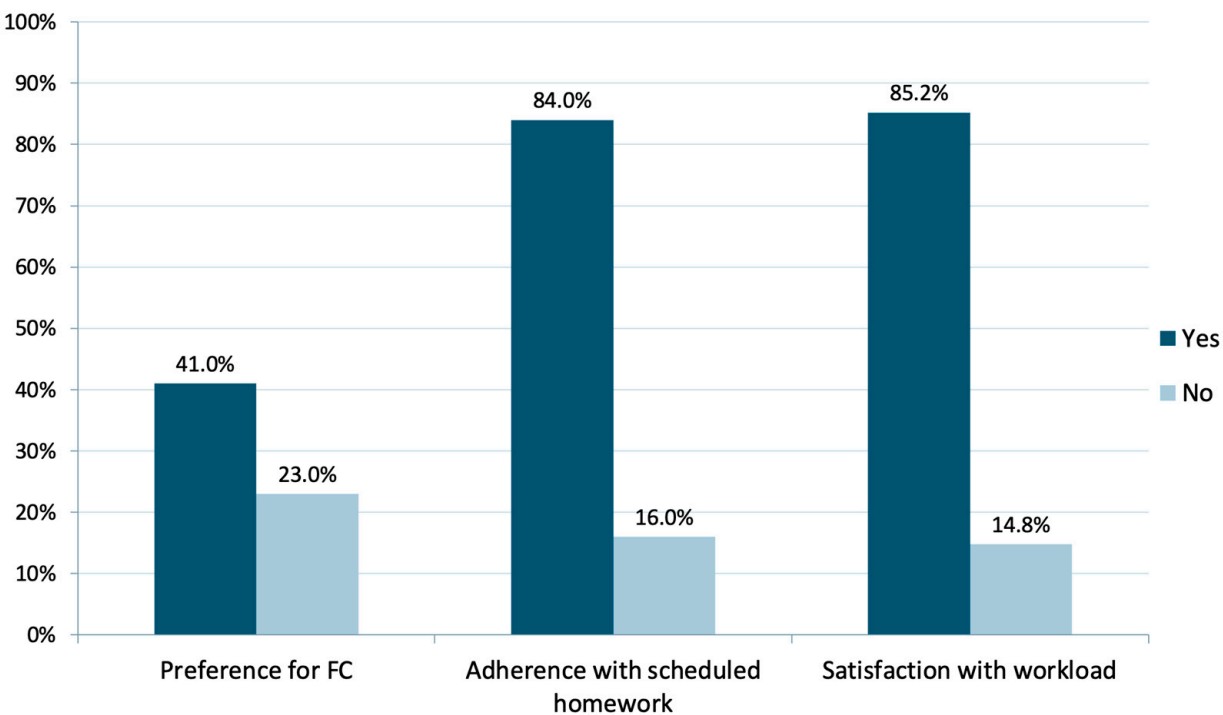

**Figure 1.** Students' preferences and satisfaction with FC.

### 3.1. Personality Traits, Locus of Control, and Teacher Evaluation

Some personality traits were associated with the scores in the teacher evaluation for the FC, as high levels of conscientiousness and high scores for internal locus of control were significantly associated with better teacher evaluation. Conscientiousness was directly associated with higher student's engagement (Rho = 0.29; $p$ = 0.002), enthusiasm (Rho = 0.23; $p$ = 0.012), and satisfaction with course organization (Rho = 0.26; $p$ = 0.004). Openness was positively associated with student's enthusiasm (Rho = 0.23; $p$ = 0.002), higher explanation clarity (Rho = 0.25; $p$ = 0.006), and larger perceived workload (Rho = 0.25; $p$ = 0.006); all correlations had a low–medium effect size (between 0.23 and 0.29) (Table 2).

**Table 2.** Correlation coefficients between course evaluation and personality of students.

| Personality Traits | Satisfaction (*p*-Value) | Engagement (*p*-Value) | Enthusiasm (*p*-Value) | Perceived Workload (*p*-Value) | Level of Self-Effort (*p*-Value) | Teacher Evaluation (*p*-Value) |
|---|---|---|---|---|---|---|
| Conscientiousness | 0.26 * (0.004) | 0.29 * (0.002) | 0.23 * (0.012) | 0.07 (0.462) | 0.24 * (0.010) | 0.12 (0.183) |
| Extraversion | 0.04 (0.693) | 0.08 (0.402) | 0.05 (0.581) | 0.06 (0.513) | 0.096 (0.299) | 0.04 (0.693) |
| Neuroticism | −0.08 (0.402) | −0.12 (0.199) | −0.02 (0.875) | −0.08 (0.370) | 0.08 (0.394) | −0.04 (0.637) |
| Openness | 0.17 (0.061) | 0.17 (0.062) | 0.23 * (0.002) | 0.25 * (0.006) | 0.08 (0.417) | 0.25 * (0.006) |
| Agreeableness | 0.14 (0.118) | 0.11 (0.217) | 0.07 (0.483) | 0.18 (0.055) | 0.24 * (0.008) | 0.01 (0.954) |

\* $p < 0.05$.

Students who preferred FC differed from those preferring the traditional methods in several personality dimensions. In this vein, students who showed a greater preference for the traditional method than for FC scored higher in neuroticism, with a medium effect size (26.5 ± 7.0 for students who preferred the traditional method vs. 22.7 ± 7.3 for students who preferred FC; t = 2.22; df = 90; $p$ = 0.022; d = 0.51). It is also noteworthy that higher

neuroticism correlated with a higher external locus of control, with a medium effect size (Rho = 0.31; $p < 0.001$), and with lower conscientiousness (Rho = $-0.26$; $p < 0.001$), also with a medium effect size.

Students with low adherence to scheduled homework tasks before classroom sessions showed higher extraversion scores (32.6 $\pm$ 4.8 for less adherent students vs. 29.6 $\pm$ 4.7 for more adherent students; t = 2.51; df = 97; medium effect size, d = 0.52; $p = 0.014$), whereas higher scores on conscientiousness were associated with a higher self-reported level of effort, with a small effect size (Rho = 0.24; $p = 0.010$).

*3.2. Final Grades*

Comparing the course in 2018–2019 with that of the previous year, the percentage of students who had not passed the subject was lower in the year when FC was administered, 35/233 (15%), than that in the previous year, 80/242 (33%) ($\chi^2$ = 21.05; df = 1; odds ratio = 2.80; $p < 0.001$).

**4. Discussion**

Our findings indicate that the FC method was highly appreciated among most students, in line with previous reports [12]. Additionally, the present study provides insights into what intrapersonal factors might influence the preference for the FC approach. We showed that students with high neuroticism and/or an externally oriented attribution style are more prone to preferring the traditional approach over the FC [16], contrary to previous reports not finding such an association [26]. Accordingly, we could not confirm that students who preferred the FC would show higher scores on conscientiousness; instead, we showed that students who prefer traditional teaching methods score higher on neuroticism, which is in turn inversely associated with variables that indicate low self-discipline, such as a higher locus of external control and lower conscientiousness, as it has been previously reported [27].

In this sense, Young-Kim (2017) showed that participants reported a high degree of personal involvement in learning, better content understanding, convenience in time and pace, and enhancement of interactions as the benefits for the FC. However, there were also some inconveniences reported by students for the FC method, in particular, students also noted that FC involves a heavy workload, a high preparation time, and a lack of familiarity with this method. The present study might nuance the pitfalls previously reported for the FC, as over 85% of the participants in the present study were satisfied with the workload delivered by the FC. In this vein, a possible source of discrepancy with other reports might be that the FC in the present study demanded a different degree of time and preparation for the students than previous work, which might favor both adherence and satisfaction for the FC in our study. As previously mentioned in [26], no differences were found between personality traits and satisfaction with the FC course. In their study [26], participants classified as having a 'Diverger learning style'—being imaginative and good at concrete experience and reflective, observation—had the lowest satisfaction with the FC method and scored higher in extraversion and conscientiousness. The authors interpreted their results in light of a previous study [28], showing that participants with high extraversion and conscientiousness are reluctant to take online classes. Thus, the discrepancies between the present study and previous ones might be related to the online component of the FC method. It is of relevance to point out that the FC delivered in the present study did not replace face-to-face courses but was restricted to modifying its methodology. Consequently, reluctance shown by participants scoring high in extraversion and conscientiousness with the FC in previous studies might be rather related to missing personal attendances, which is not applicable to our results. Moreover, studies referring to learning styles must be taken into consideration with considerable caution, as the existence of so-called 'learning styles' has been put into question for not having appropriate scientific support and is even considered a myth [29].

Other interpersonal factors might not favor a student-centered approach; in this vein, students who scored high in neuroticism preferred traditional lessons, and students with high extraversion levels showed lower levels of FC adherence. Our results align well with those of Zhang [30] in that open, conscientious, and emotionally stable participants are more prone to engaging in deep interactive learning processes. Our study aligns partially with those reporting that academic achievement and class attendance (as a measure of invested effort) are related to variables associated with better self-discipline [17,18]. The present results highlight that conscientiousness is closely related to higher course engagement, enthusiasm, and course organization and to higher levels of self-effort.

In light of the present results, teachers should take into consideration the possible reservations that students might have about new teaching methodologies related to core personality traits.

Similar to previous studies, our results indicate that the use of an FC in nursing education could increase students' academic performance [12]. We also show that other noncognitive factors can influence nursing students' teaching methodology preferences and academic results [23].

Our study has several limitations. Firstly, we used a sample of students from only one institution, limiting the results' generalizability. Secondly, we could not compare the main variables with the previous course since the outcomes were unavailable. The main strengths of the present study include the combined assessment of student satisfaction, personality traits, and attributional styles, together with self-perceived effort and student grades within nursing students in the context of a growing teaching methodology.

Further studies should evaluate how the FC methodology prepares nurses in understanding and analyzing statistical data that may be useful for clinical practice.

## 5. Conclusions

The present study found that the FC approach was valuable and suitable for a biostatistics course and was positively rated by students. Most of the students preferred FC to traditional lectures, except those who scored high on neuroticism, preferring traditional classes instead. Personality traits and locus of control were, thus, associated with preferences about a given educational methodology (FC vs. traditional lectures), as well as with self-reported effort and adherence to the FC methodology. The FC methodology may facilitate the acquisition of relevant skills in biostatistics by promoting adherence and by improving learning outcomes, especially in students with a high tendency to be careful and efficient. We conclude that conscientiousness plays an important role in adherence, satisfaction, and performance with FC. In summary, the tendency toward self-control, organization, motivation, and reliability constitutes key intrapersonal factors that influence the preference for non-traditional methods of teaching.

**Author Contributions:** Conceptualization, A.M.Y. and M.B.-V.; methodology, A.M.Y., D.A.-R. and M.B.-V.; formal analysis, A.M.Y., D.A.-R. and M.B.-V.; investigation, A.M.Y.; data curation, A.M.Y., D.A.-R. and M.B.-V.; writing—original draft preparation, A.M.Y. and M.B.-V.; writing—review and editing, D.A.-R. All authors have read and agreed to the published version of the manuscript.

**Funding:** This research received no external funding.

**Institutional Review Board Statement:** This project was approved by the Ethics Research Committee (CER) of the University of the Balearic Islands (CER76/18).

**Informed Consent Statement:** Informed consent was obtained from all subjects involved in the study.

**Data Availability Statement:** Derived data supporting the findings of this study are available from the corresponding author on request.

**Acknowledgments:** The authors would like to thank all students who voluntarily participated in this study.

**Conflicts of Interest:** The authors declare no conflict of interest.

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
