# Peer review of "Personality, Preferences, Satisfaction, and Achievement in a Biostatistics Course: Traditional versus Flipped Classrooms in Nursing Education"

_education, doi:10.3390/educsci13020197_

Round 1

Reviewer 1 Report

This study could contribute to the literature in terms of the dependent variables included to research questions. Howvever, there are major concerns in the manuscript and it needs to be improved:

1. the literature is lack of explanation of personal traits, satisfaction, locus of control and satisfaction studies, how these terms are defined, how they effect to student learning and why these variables can be important in flipped classrooms approach.

2. Similarly the authors must explain other individual factors that might have an effect on student learning in flipped classroms and explain why the dependent variables were selected for this study

3. the research questions are not clearly stated. Please add them with in a numbered items.

4. In the method section, please expain in a table how the flipped and conventional version of this course was given. What kind of changes or similarities were applied

5. the study does not apply an experimental design for the same cohort. It compares the flipped classroom application with the conventional one (previous cohort). However, it is not clear if the groups, learning objectives, learning tasks, assignments given were equal for each cohort. That might be a bias for the flipped classroom approach.

6. please re-orginize the results sections based on the research questions.

7. In table 1, there are no meaningful correlations for the two variables. there are also low correelations for the three variables. the result show that personality of students did not effect (or have very lof effect) on course evaluation. There might be problems for measurement tools or evaluation tool.

8. final grades of the students were compared but there isno evidence for the equivalence of the final exams. please present test statistics and item statistics to show the exams were equivalent to each other.

9. Failed students were higher in flipped classrooms group. there might be two reasons for this results: 1) flipped classroom group exam was more difficult 2) Too many learning tasks were given to the flipped classroom and they could not deal with hese tasks and as a consequense, learning gains were lower in flipped group than the traidional group.

10. the discussion section focused on learning styles however this study is not directly related to learning styles thory. There is no discussion for the results obtained.

Author Response

Reviewer 1:

This study could contribute to the literature in terms of the dependent variables included to research questions. However, there are major concerns in the manuscript, and it needs to be improved:

  1. The literature is lack of explanation of personal traits, satisfaction, locus of control and satisfaction studies, how these terms are defined, how they effect to student learning and why these variables can be important in flipped classrooms approach

Response to comment#1: We thank the Reviewer for making us notice that the former version of the manuscript needed additional explanation about personal traits, satisfaction and locus of control, together with their mutual associations with student learning and their potential relevance for the FC approach. We have now introduced a brief description of such terms.

Page 2, introduction section (lines 61-69): ‘In this vein, previous studies have highlighted that class attendance is associated to variables that reflect high levels of self-discipline (e.g., high conscientiousness), a sense of internal control (internal locus of control) over academic achievement. Many of these variables also show significant relationships with academic outcomes (Lievens et al., 2002; Robbins et al., 2004). More specifically, locus of control refers to the explanations that people provide for the causes of their behaviors, and the psychological terms aiming to explain these processes are referred to as attributions. In psychology, these attributions help to better understand human motivation and sense of competence.

We have also added a short description on conscientiousness and its relationship with locus of control, in Page 2 (lines 73-83), which read: ‘Additionally, an internal locus of control tends to be directly associated with some dimensions of personality, such as conscientiousness (Morrison, 1997), which is one of the most powerful non-cognitive predictors of academic achievement (Dumfart & Neubauer, 2016). In this regard, we departed from the Big Five model of personality (see [19]), which includes the dimensions neuroticism (emotional instability), extraversion, conscientiousness, agreeableness and openness to experience. We were particularly interested (but not exclusively) in exploring the role of conscientiousness, which is the tendency to do a task efficiently and carefully, and to take obligations seriously. People scoring high in conscientiousness tend be organized and tend to show behaviors of self-discipline and aim for achievement (see Roberts et al., 2009)’.

We also want to highlight that the Introduction section aims to recapitulate the main rationale of introducing both individual factors (page 4, lines 166-167): ‘Thus, we aimed to consider both potentially relevant variables regarding the FC model since both seem to play a role with discipline and commitment’.  

New references:

Dumfart, B., & Neubauer, A. C. (2016). Conscientiousness is the most powerful noncognitive predictor of school achievement in adolescents. Journal of Individual Differences, 37(1), 8–15. https://doi.org/10.1027/1614-0001/a000182

Lievens, F., Coetsier, P., De Fruyt, F., & De Maeseneer, J. (2002). Medical students’ personality characteristics and academic performance: A five-factor model perspective. Medical Education, 36, 1050–1056.

Morrison, K. A. (1997). Personality Correlates of the Five-Factor Model for a Sample of Business Owners/Managers: Associations with Scores on Self-Monitoring, Type a Behavior, Locus of Control, and Subjective Well-Being. Psychological Reports80(1), 255–272. https://doi.org/10.2466/pr0.1997.80.1.255

Robbins, S. B., Lauver, K., Le, H., Davis, D., Langley, R., & Carlstrom, A. (2004). Do psychosocial and study skill factors predict college outcomes? A meta-analysis. Psychological Bulletin, 130, 261–288.

Roberts, B. W., Jackson, J. J., Fayard, J. V., Edmonds, G., & Meints, J. (2009). Conscientiousness. In M. R. Leary & R. H. Hoyle (Eds.), Handbook of individual differences in social behavior (pp. 369–381). The Guilford Press.

  1. Similarly the authors must explain other individual factors that might have an effect on student learning in flipped classrooms and explain why the dependent variables were selected for this study

Response to comment#2: As stated in the previous response, we were inclined to study individual factors that might tap on self-discipline and commitment, such as the internal locus of control and conscientiousness, both being terms related to the field of personality. As we have noted, we expected that positive perceptions on the FC methodology would be associated to variables that have been related with high class attendance. Such candidates were high levels of self-discipline (e.g., high conscientiousness), and a sense of internal control (internal locus of control) over academic achievement, given their positive role on academic outcomes (see Lievens et al., 2002; Robbins et al., 2004).

As the Reviewers points out, it is highly plausible that other individual factors other than locus of control or personality also play a role on student learning in the FC setting (for instance, preferences for technology or general preferences for a given teaching method, such as lab classes, standard lectures and/or discussion groups, among others; see Chamorro-Premuzic, Furnham & Lewis, 2008). However, the study of a wider plethora of individual factors that influence on learning outcomes in FC fell outside the aims of the present work, and we do not have such data available. However, we believe that future works would greatly benefit from the inclusion of additional relevant factors that add significant amounts of explained variance on learning outcomes in the FC setting.

Reference:

Chamorro-Premuzic, T., Furnham, A., & Lewis, M. (2007). Personality and approaches to learning predict preference for different teaching methods. Learning and Individual Differences, 17 (3), 241-250. https://doi.org/10.1016/j.lindif.2006.12.001

  1. The research questions are not clearly stated. Please add them with in a numbered items.

Response to comment#3: Thank you very much. According to your comment we have added numbered items in the research questions (page 3, lines 99-103).

  1. In the method section, please explain in a table how the flipped and conventional version of this course was given. What kind of changes or similarities were applied

Response to comment#5: Thank you very much for your suggestion. Following you recommendation we have added a comparative table between flipped and conventional version of biostatistics course.

  1. The study does not apply an experimental design for the same cohort. It compares the flipped classroom application with the conventional one (previous cohort). However, it is not clear if the groups, learning objectives, learning tasks, assignments given were equal for each cohort. That might be a bias for the flipped classroom approach.

Response to comment#6: Thank you for pointing this out. We agree with this comment. We have included in the methods section, data analysis (page 4, lines 163-166): “Finally, after the final examination, students’ marks for the 2018-19 current academic course were compared to those of the previous course (2017-2018), retrieved using computerized registers. Group characteristics, learning objectives and evaluation were similar for both cohorts although learning task were different (Table 1)”.

  1. please re-organize the results sections based on the research questions.

Response to comment#7: We appreciate the Reviewer’s comment in this regard. Accordingly, the results are now organized based on research questions order.

  1. In table 1, there are no meaningful correlations for the two variables. there are also low correlations for the three variables. the result show that personality of students did not effect (or have very low effect) on course evaluation. There might be problems for measurement tools or evaluation tool.

Response to comment#7: We agree that the correlations are low but statistically significant. It is worth noting that we used a self-administered validated personality questionnaire (Neo-FFI). However, these associations were not very high but significant.

  1. final grades of the students were compared but there Is no evidence for the equivalence of the final exams. Please present test statistics and item statistics to show the exams were equivalent to each other.

Response to comment#8: Both final exams were similar because were based in about the same   problem solving (e.g. changing only the figures values or simple change of wording or the name of the variables).

  1. Failed students were higher in flipped classrooms group. there might be two reasons for this results: 1) flipped classroom group exam was more difficult 2) Too many learning tasks were given to the flipped classroom and they could not deal with these tasks and as a consequence, learning gains were lower in flipped group than the traditional group.

Response to comment#9: We apologize because the wording was not clear. In this regard, students in the FC group failed less than students in the traditional classroom. We have changed the wording to clarify this aspect.

  1. the discussion section focused on learning styles however this study is not directly related to learning styles theory. There is no discussion for the results obtained.

Response to comment#10: We agree that learning styles was not the focus of the present study and thus, the discussion section should be addressed to the main results and predictions presented earlier in the manuscript. In the present version of the paper, we have aimed to solve this. For instance, we have added the following paragraph in the discussion section, which is closely related to the main aims and variables studied in terms of individual factors:

Page 6, lines 233-238: ‘Accordingly, we could not confirm that students who preferred the FC scored higher on conscientiousness; instead, we show that students who prefer the traditional teaching methods score higher on neuroticism, which is in turn inversely associated to variables that indicate low self-discipline, such as a higher locus of external control and lower conscientiousness, as it has been previously reported (see Beckman, Wood & Minsbashian, 2010)’.

Page 7, lines 271-256: ‘Our study aligns partially with those reporting that academic achievement and class attendance (as a measure of invested effort) is related to variables associated with larger self-discipline, (Lievens et al., 2002; Robbins et al., 2004). The present results highlights that conscientiousness is closely related with higher course engagement, enthusiasm and course organization and larger levels of self-effort’.

Reference:

Beckmann, N., Wood, R.E., Minbashian, A. (2010). It depends how you look at it: On the relationship between neuroticism and conscientiousness at the within- and the between-person levels of analysis. Journal of Research in Personality, 44, (5), 593-601. https://doi.org/10.1016/j.jrp.2010.07.004.

Reviewer 2 Report

It might be published after minor revision.

In fact, the text is ready to be published in its present form.

Before forwarding their text to the editor, Authors may decide:

¾     Make minimal improvements (abstract, conclusions); the text will be ready for publishing but may cause some critics.

¾     Invest some effort and improve the text; the text will be ready for publishing.

¾     Invest some more effort in reconstructing indicated issues. The text will be a high-quality publication.

Please refer to the appended file, where:

A list of strengths is provided.

The list of defects is formulated.

Suggestions for improvements are given.

Author Response

Reviewer 2:

It was a pleasure having the opportunity to review this manuscript. The authors describe an effort to identify the specific factors influencing personal preferences for teaching methodology and academic outcomes in biostatistics among nursing students. They tried to evaluate the associations between personality traits and priorities for theflipped classroom or traditional lessons. The context of the investigation involves satisfaction and achievement in the subject of biostatistics in nursing education. The authors found that most students preferred flipped classrooms to traditional lectures. As a result of introducing the flipped classroom teaching technique, the percentage ofstudents who did not pass the biostatistics exam was reduced by 50%. The author concluded that flippedclassroom was adequate for the biostatistics subject. They recommend that teachers should reconsider possible reserves about new teaching methods, which can be related to student personality. After carefully reviewing this manuscript, I find that some areas need to be addressed before the paper is ready for publication. The formulated suggestions are meant to support the further development of the manuscript. In fact, the text is ready to be published in its current form. Before forwarding their text to the editor, Authors may decide:

  • Make minimal improvements (abstract, conclusions); the text will be ready for publishing but may cause some critics.
  • Invest some effort and improve the text; the text will be ready for publishing.
  • Invest some more effort in reconstructing indicated issues. The text will be a high-quality publication.

Some aspects should be improved to avoid misunderstanding and lack of clarity and enhance the publication’s overall quality. Minor modifications consisted of rewriting an Abstract; therefore, the introduction and the new version of the Conclusions would radically improve the text quality. The authors are recognised experts on the topic. They should remember that the text is meant not only for equally expert readers.

General thoughts. Strengths.

1.Interesting Report. Has potential for publication.

2.The study takes up a significant socio-economic problem.

3.The report contains information on methodological work and the case study experiment. The originality consists in the expert description of the problems connected with the implementation of known research techniques to unexplored (enough) new socio-economic phenomena, which seems to be an interestingly selected area of the analysis. The uniqueness of the approach consists of a comprehensive description of a specific socio-economic problem. The subject matter is so far insufficiently described.

4.The study’s scientific value is based on the use of well-known, descriptive and analytical tools for highlighting the vital socio-economic area. Some methodological contribution is provided.

5.The scientific level allows for generalisation on other geographical, cultural and organisational areas.

6.The report does have a correct and complete structure. The text division (headings) is chosen due to (implicit) justification.

7.The authors provided a sufficiently extensive, comprehensive review of the newest world literature on the topic.

8.The scope is sufficiently described and justified. The authors did not address the broader generalisation potential should be. A more comprehensive international and interregional background should be included. Are the analysed problems unique in how they profit from the proposed analytical approach?

9.The reviewer makes no reservations about the justification for the choice of topic and scope (except for the lack of broader national and international background).

10.Authors skilfully used relatively advanced analytical tools.

11.The descriptive part of the research may be accepted. Included elements of ANALYSIS enrich the results.

12.The language is on a satisfactory level. It may (and should) be improved

Response: We are very grateful to the reviewer for their encouragement. Thank you ever so much for all your comments.

General thoughts. The work contains defects.

1.The severe weakness of the text. The goal is NOT specified explicitly enough. Authors should explain why the document was prepared and who should profit from the new knowledge acquired in their investigations. Unfortunately, the Authors fail to specify WHICH TYPE OF BENEFITS and WHAT TYPE OF STAKEHOLDERS. The description of their inquiry results may be of interest or support the managerial and policy decisions.

Authors inform: This study sought to evaluate the associations between personality traits and preferences for the flipped classroom or traditional lessons, satisfaction, and achievement in the subject of biostatistics in nursing education The evaluation (…) can never be a task (goal). There has to be the reason why the research is necessary. Therefore, EVALUATION can only be a tool to achieve the goal. Usually, authors propose analytical tools to pursue some theoretical, cognitive, methodological, empirical or practical (implemental) goal.The authors did not (explicitly) formulate such tasks.

Response: Thank you for pointing this out. We have reformulated the goals “To address the following research questions: 1) How the big five personality traits and locus of control are associated with satisfaction, academic achievement, and preference for a given teaching methodology? 2) How is the locus of control associated with satisfaction, academic achievement, and preference for a given teaching methodology? 3) What is the impact of flipped class in academic performance of students?” (page 2, lines 99-103).

Furthermore, we have added the reason why the research is necessary “A recent systematic review reported inconsistent associations between individual (non-cognitive) factors and academic outcomes among students of health professions; however, higher conscientiousness, academic resilience, and grit were generally associated with better outcomes [18]. Nevertheless, there is a knowledge gap about how individual factors, such as locus of control or personality dimensions, are associated with personal preferences for a given teaching methodology and academic outcomes among nursing students. Thus, we aimed to consider both potentially relevant variables regarding the FC model since both seem to play a role with discipline and commitment” (page 2, lines 84-92).

2.The weakness of the text. There is a lack of data description. The authors fail to describe the respondents. The demographic and socioeconomic characteristics are missing.

Generally speaking, authors should elaborate description of the data collection process and data set composition.

Response: Thank you very much. Unfortunately, the main variables that we collected were age, sex, race and previous academic level that are already reported: “The mean age was 22.4±7 years, 93.4% of participants were Caucasian, and 79.6% were women. Most students were upper secondary education (49%) and advanced vocational training (34%) graduates who passed the university admission exam; the remaining students accessed through the special access for people over 25 years (9%), as second-degree students or through a university degree change (8%)”.

3.The weakness of the text. The authors do not provide enough literature references concerning similar data analysis. This fact, combined with vague, imprecise statements about the analytical techniques used to study the collected data, makes it difficult to assess whether the methods used for data analysis are appropriate. The lack of precision and literature references prevents further development of the proposed approach in a situation where other authors would try to improve or even test the applicability of the described construction. The idea of the locus of control measurement (LOC) has a long tradition in Psychology. Numerous authors designed various variants of the instrument. The problem of the dimensionality of the construct still is unclear. The controversy on the issue is unsolved. The authors should provide some insight into the directions of the discussion.

The authors provided a review of the world literature on the topic. It is not clear which inference contained in the report results is from the literature query. Which recommendations for the current report come from the surveyed literature? It seems desirable that the analytical part of the text should have elements of generalising style. Authors may try and enrich recommendations.

Response: Thank you very much. We thank the Reviewer for making us notice that the former version of the manuscript needed additional explanation about personal traits, satisfaction and locus of control, together with their mutual associations with student learning and their potential relevance for the FC approach. We have now introduced a brief description of such terms.

Page 2, introduction section (lines 61-69): ‘In this vein, previous studies have highlighted that class attendance is associated to variables that reflect high levels of self-discipline (e.g., high conscientiousness), a sense of internal control (internal locus of control) over academic achievement. Many of these variables also show significant relationships with academic outcomes (Lievens et al., 2002; Robbins et al., 2004). More specifically, locus of control refers to the explanations that people provide for the causes of their behaviors, and the psychological terms aiming to explain these processes are referred to as attributions. In psychology, these attributions help to better understand human motivation and sense of competence.

We have also added a short description on conscientiousness and its relation with locus of control, in Page 2 (lines 73-83), which read: ‘Additionally, an internal locus of control tends to be directly associated with some dimensions of personality, such as conscientiousness (Morrison, 1997), which is one of the most powerful non-cognitive predictors of academic achievement (Dumfart & Neubauer, 2016). In this regard, we departed from the Big Five model of personality (see [19]), which includes the dimensions neuroticism (emotional instability), extraversion, conscientiousness, agreeableness and openness to experience. We were particularly interested (but not exclusively) in exploring the role of conscientiousness, which is the tendency to do a task efficiently and carefully, and to take obligations seriously. People scoring high in conscientiousness tend be organized and tend to show behaviors of self-discipline and aim for achievement (see Roberts et al., 2009)’.

References:

Dumfart, B., & Neubauer, A. C. (2016). Conscientiousness is the most powerful noncognitive predictor of school achievement in adolescents. Journal of Individual Differences, 37(1), 8–15. https://doi.org/10.1027/1614-0001/a000182

Lievens, F., Coetsier, P., De Fruyt, F., & De Maeseneer, J. (2002). Medical students’ personality characteristics and academic performance: A five-factor model perspective. Medical Education, 36, 1050–1056.

Morrison, K. A. (1997). Personality Correlates of the Five-Factor Model for a Sample of Business Owners/Managers: Associations with Scores on Self-Monitoring, Type a Behavior, Locus of Control, and Subjective Well-Being. Psychological Reports80(1), 255–272. https://doi.org/10.2466/pr0.1997.80.1.255

Robbins, S. B., Lauver, K., Le, H., Davis, D., Langley, R., & Carlstrom, A. (2004). Do psychosocial and study skill factors predict college outcomes? A meta-analysis. Psychological Bulletin, 130, 261–288.

Roberts, B. W., Jackson, J. J., Fayard, J. V., Edmonds, G., & Meints, J. (2009). Conscientiousness. In M. R. Leary & R. H. Hoyle (Eds.), Handbook of individual differences in social behavior (pp. 369–381). The Guilford Press.

4.The weakness of the text. Research techniques were applied to register several practical (implemental) aspects separately. The scientific level is lacking in the discussion on generalization aspects. It is unclear which statements are specific to the surveyed population, students and teachers of the analysed school. There is a lack of clarity on whether the comments refer exclusively to surveyed respondents or have a broader meaning. It would be interesting for the wider public to know which general recommendations concern any society (culture, geography, language group, education system, etc The authors should clarify.The analytical part needs to be rewritten. The revised text should contain a description and justification of the Authors’ position along with the Authors’ assessments’ of the following:

4.1.The theoretical and practical meaning of individual approaches that exist in the literature,

4.2.The authors’ position towards the importance of the individual, theoretical frameworks for the general (world, language group, etc.) and local policy and organisation (surveyed school).

4.3.Authors should introduce an attempt to classify theoretical frameworks existing in world literature. It would enrich the analysis results message for the HE service policy and practitioners in institutions of different types and levels, along with potential users’ characterisation.

5.The abstract and introduction should be modified.

5.1.The research problem and research goals were NOT identified in the work.

5.2.The selection of the theoretical basis of the research was NOT appropriately described and justified.

6.The selection of statistical and econometric techniques was NOT appropriately justified and described. Therefore, the methods of literature selection are not precisely explained

7.The text lacks elements of (interregional, national and international) comparisons, in-depth discussion of broader reasons (sources) and consequences of identified developments. The description refers EXCLUSIVELY to the individual, selected one after one, types of attitude. It is not a standard approach in scientific research. It should have generalisation potential and not purely case(s) study character.

8.Authors should try and distinguish which statements are the Authors’ opinion, the literature knowledge, and the analysis outcome. The used literature references are (almost exclusively) referred to in such a manner that it is not clear why the publication is cited. Usually, there are no details on whether the individual mentioned authors support the Authors’ theses and findings. Authors should precisely specify which references support their position and why and which are in opposition to their conclusions. There is no generalisation effort in the literature review.

Authors should reformulate the text of the literature review.

Instead of being purely reporting and descriptive, the text’s style needs to be analytical, with generalising indications.

9.Authors use statistical a designed for metric scale. Statistical software, however smart, is not able to distinguish whether provided data is metric or is coming from the weaker measurement scale. Authors should consult experts on whether their information suits the selected quantitative analytical tools.

10.The severe weakness of the text. In part entitled CONCLUSIONS, the conclusions are NOT included. It is merely a description of the submission text. The authors repeated analogous illustrations from the introductory parts of the report.

The formulation of conclusions should contain elements that are anchored in research findings. They should be based on analytical statements describing the MERIT TOPIC analysis results.

Response: We agree that discussion section should be addressed to the main results and predictions presented earlier in the manuscript. In the present version of the paper, we have aimed to solve this. For instance, we have added the following paragraph in the discussion section, which is closely related to the main aims and variables studied in terms of individual factors:

Page 6, lines 233-238: Accordingly, we could not confirm that students who preferred the FC scored higher on conscientiousness; instead, we show that students who prefer the traditional teaching methods score higher on neuroticism, which is in turn inversely associated to variables that indicate low self-discipline, such as a higher locus of external control and lower conscientiousness, as it has been previously reported (see Beckman, Wood & Minsbashian, 2010)’.

Page 7, lines 271-256: ‘Our study aligns partially with those reporting that academic achievement and class attendance (as a measure of invested effort) is related to variables associated with larger self-discipline, (Lievens et al., 2002; Robbins et al., 2004). The present results highlights that conscientiousness is closely related with higher course engagement, enthusiasm and course organization and larger levels of self-effort’.

Reference:

Beckmann, N., Wood, R.E., Minbashian, A. (2010). It depends how you look at it: On the relationship between neuroticism and conscientiousness at the within- and the between-person levels of analysis. Journal of Research in Personality, 44, (5), 593-601. https://doi.org/10.1016/j.jrp.2010.07.004.

DETAILED REMARKS

1.The abstract should be modified. The abstract should contain parts:

1.1.Purpose of the publication; GOAL;

1.2.Literature review (state of the art) indication;

1.3.Design/methodology/approach indication;

1.4.Main findings;

1.5.Conclusions. Summary of results, research implications and limitations; the policy recommendations;

1.6.Keywords;

The requirement for the abstract and its structure is formulated in the journal template for authors. Authors should refer there.

2.Data analysis.

2.1.The authors use statistical and econometric techniques designed for metric scale. However smart statistical software cannot distinguish whether the data is measured on a strong, metric scale (ratio or interval) or is coming from the weaker, nonmetric measurement scale (ordinal or nominal). The authors should consult an expert, whether their information suits the selected quantitative analytical tools. Most of the techniques are designed for metric data, and the measurement results seem nonmetric in the analysed dataset. The reviewer formulates serious doubt whether the solemn statistical analysis is possible without choosing dedicated techniques for nonmetric data with available data type.

2.2.The interesting information comes out when using quantiles (median, quartiles, quintiles, deciles, percentiles). Authors may check and see whether additional knowledge value comes with such indicators.

3.Editing

3.1.The star: 0.26*.

The meaning of the star in table 1 is not explained.

Response: Sorry for this mistake, we have added in the table.

4.Standards

A rule (standard) recommends not to give percentage points values for quantities lower than one hundred items.

5.The propaganda.

5.1.Analyses were performed using the Statistical Package for the Social Sciences (SPSS) soft-139 ware version 25.0 (IBM Company, New York, NY, USA) for Windows. Statistical tests 140 were two-sided, and p values <0.05 were considered statistically significant.

It looks like an advertisement for an IBM product.

The tools for data analysis are not an SPSS idea. SPSS contains procedures known from the literature. With equal success, the authors could use different packages, Statistica, PCGive; Stata, procedures on R-CRAN platform, GRETL, etc.

Maybe it would be more advisable to reference the statistics textbook to describe the data analytical techniques.

6.The personification of inanimate phenomena

6.1.FC allows students (…)

6.2.Our findings indicate (…)

6.3.systematic review reported (…)

6.4.the literature supports (…). Literature is a research tool; the literature cannot support it. People can support them. The sentnce should be modified, e.g. We (authors) support (some findings).

6.5.The present study found (…)

6.6.and many other examples of such personification.

7.Authors did not formulate Conclusions that are anchored in the research results. The text of the part contains NO conclusions. The text should be added with elements that are anchored in research findings.

The text should contain analytical statements describing the MERIT TOPIC. The further research recommendation should (might) be included.

7.1.The possible solution for the Conclusions part is dividing it into three parts: parts: Conclusions resulting from the STATE OF ART analysis,

Findings resulting from own research, but only from the study described in the current text, Policy implications,

Further research directions.

7.2.Conclusions should be fully anchored in the methodological proposals and analysis results drawn from the empirical material. The last part should be formulated based on the GOAL formulation, i.e.

7.2.1.WHY the findings are essential;

7.2.2.for WHOM and

7.2.3.WHAT FOR; for which type of policy or for what kind of decisions.

7.2.4.Statements may be complemented with the information on

7.2.5.HOW the results were obtained and how the results may be implemented in the education services

R: Thank you very much. We are not sure about some of your statements, but we have modified the manuscript in order to clarify the main goals of the study, introduction, methods and discussion sections.
